# Light-Activated Liposomes Coated with Hyaluronic Acid as a Potential Drug Delivery System

**DOI:** 10.3390/pharmaceutics12080763

**Published:** 2020-08-12

**Authors:** Otto K. Kari, Shirin Tavakoli, Petteri Parkkila, Simone Baan, Roosa Savolainen, Teemu Ruoslahti, Niklas G. Johansson, Joseph Ndika, Harri Alenius, Tapani Viitala, Arto Urtti, Tatu Lajunen

**Affiliations:** 1Drug Research Program, Division of Pharmaceutical Biosciences, Faculty of Pharmacy, University of Helsinki, Viikinkaari 5 E, FI-00790 Helsinki, Finland; otto.kari@helsinki.fi (O.K.K.); shirin.tavakoli@helsinki.fi (S.T.); petteri.parkkila@helsinki.fi (P.P.); s.d.baan@uu.nl (S.B.); roosa.pp.savolainen@helsinki.fi (R.S.); teemu.ruoslahti@aalto.fi (T.R.); tapani.viitala@helsinki.fi (T.V.); arto.urtti@helsinki.fi (A.U.); 2Pharmaceutics, Utrecht Institute for Pharmaceutical Sciences, Utrecht University, P.O. Box 80.082, 3508 TB Utrecht, The Netherlands; 3Drug Research Program, Division of Pharmaceutical Chemistry and Technology, Faculty of Pharmacy, University of Helsinki, Viikinkaari 5 E, FI-00790 Helsinki, Finland; niklas.johansson@helsinki.fi; 4Human Microbiome Research, Faculty of Medicine, University of Helsinki, Haartmaninkatu 3, FI-00290 Helsinki, Finland; joseph.ndika@helsinki.fi (J.N.); harri.alenius@helsinki.fi (H.A.); 5Institute of Environmental Medicine, Karolinska Institutet, 171 77 Stockholm, Sweden; 6School of Pharmacy, Faculty of Health Sciences, University of Eastern Finland, Yliopistonranta 1, 70210 Kuopio, Finland; 7Institute of Chemistry, St. Petersburg State University, Petergof, Universitetskii pr. 26, 198504 St. Petersburg, Russia; 8Laboratory of Pharmaceutical Technology, Department of Pharmaceutical Science, Tokyo University of Pharmacy & Life Sciences, 1432-1 Hachioji, Tokyo 192-0392, Japan

**Keywords:** hyaluronic acid, liposome, drug release, light activation, stability, mobility, biocorona

## Abstract

Light-activated liposomes permit site and time-specific drug delivery to ocular and systemic targets. We combined a light activation technology based on indocyanine green with a hyaluronic acid (HA) coating by synthesizing HA–lipid conjugates. HA is an endogenous vitreal polysaccharide and a potential targeting moiety to cluster of differentiation 44 (CD44)-expressing cells. Light-activated drug release from 100 nm HA-coated liposomes was functional in buffer, plasma, and vitreous samples. The HA-coating improved stability in plasma compared to polyethylene glycol (PEG)-coated liposomes. Liposomal protein coronas on HA- and PEG-coated liposomes after dynamic exposure to undiluted human plasma and porcine vitreous samples were hydrophilic and negatively charged, thicker in plasma (~5 nm hard, ~10 nm soft coronas) than in vitreous (~2 nm hard, ~3 nm soft coronas) samples. Their compositions were dependent on liposome formulation and surface charge in plasma but not in vitreous samples. Compared to the PEG coating, the HA-coated liposomes bound more proteins in vitreous samples and enriched proteins related to collagen interactions, possibly explaining their slightly reduced vitreal mobility. The properties of the most abundant proteins did not correlate with liposome size or charge, but included proteins with surfactant and immune system functions in plasma and vitreous samples. The HA-coated light-activated liposomes are a functional and promising alternative for intravenous and ocular drug delivery.

## 1. Introduction

Liposomes are a well-accepted drug delivery system and still the most common nanomedicines in clinical trials [1]. Most liposomal carriers are coated with polyethylene glycol (PEG) to prolong their residence times in blood circulation [2,3], but there are still problems related to reduced uptake into the target cells [3,4], immunogenicity [5,6], and undesired protein interactions [7]. Therefore, alternatives for PEG were proposed, including polyvinylpyrrolidone (PVP) [8], polycarbonates [9], pullulan [10], polyoxazolines [11], and hyaluronic acid (HA) or hyaluronan [12].

HA is a negatively charged natural polysaccharide of d-glucuronic acid and *N*-acetyl-d-glucosamine disaccharide units that are connected via alternating β-1,4 and β-1,3 glycosidic bonds. HA contains carboxyl, hydroxyl, and amino groups that can be derivatized. HA is biocompatible, biodegradable, non-toxic, and less immunogenic than PEG [12,13]. In addition, HA binds the CD44 receptors that are overexpressed in melanomas, lymphomas, breast cancer, lung, and rectal tumor cells [12] and present in the retinal pigment epithelium [14]. Thus, the HA coating combines steric stabilization (“stealth”) and active targeting functions. Furthermore, HA improves liposomal stability in freeze-drying due to its high water-binding capacity [15]. 

Passive drug release from liposomes is often erratic and inadequate [16], but this can be improved by external triggers such as light [17,18]. Light activation is feasible for drug release in superficial organs (eye and skin), as well as in deeper tissues with light guides, and light parameters can be optimized for treatment (e.g., wavelength, light intensity, exposure time, and beam size) [19]. We developed light-activated PEG-coated liposomes using indocyanine green (ICG) as the light-sensitizing compound [20,21,22]. Light-activated release of small and large compounds was achieved with near-infrared (NIR) light with only five-second light exposures [22,23,24]. Amphiphilic ICG can be integrated into the liposomal bilayers or clustered with the hydrophilic coating on the liposome surface (e.g., PEG). ICG is a fluorescent dye approved by the European Medicines Agency (EMA) [23] and the United States (US) Food and Drug Administration (FDA) [24] for angiographic and lymphatic system imaging in the clinics [25]. 

The acquired protein corona determines the “biological identity” and fate of liposomes in a biological environment [26,27]. We demonstrated that ICG clustered with PEG on the liposome surface influenced the structure and composition of the protein corona, including the beneficial enrichment of “dysopsonin” molecules such as clusterin and apolipoprotein E [28]. Even though the liposomal protein coronas were characterized in plasma, their formation in the vitreous humor of the eye was studied only in our recent publication [29]. The structure and contents of the vitreal liposome corona were not dependent on the composition of 50 nm anionic light-activated liposomes [28]. This is in line with earlier observations with silica and gold nanoparticles in dilute dog vitreous samples [30]. The slight increase (10–12%) in liposomal diameter due to corona formation did not affect the mobility of the liposomes in the vitreous humor [29].

HA is an endogenous component in the vitreous humor and, in that respect, an interesting coating material for ocular liposomes. This report describes the development of light-activated liposomes with HA coating. We produced HA–lipid conjugates in-house and characterized the HA-coated liposomes for drug release, stability, protein corona formation, and mobility in the vitreous humor.

## 2. Materials and Methods 

### 2.1. Materials

Dipalmitoylphosphatidylcholine (DPPC), 1,2-distearoyl-*sn*-glycero-3-phosphocholine (DSPC), lysophosphatidylcholine (LysoPC), 1,2-distearoyl-*sn*-glycero-3-phosphorylethanolamine (DSPE), 1,2-distearoyl-*sn*-glycero-3-phosphoethanolamine-*N*-[amino(polyethylene glycol)-2000] (DSPE-PEG), 1,2-distearoyl-*sn*-glycero-3-phosphoglycerol (DSPG), and the 1,2-dioleoyl-*sn*-glycero-3-phosphoethanolamine-*N*-(Cyanine 5) lipid dye were purchased from Avanti Polar Lipids, Inc. (Alabaster, AL, USA). Citrate-phosphate-dextrose (CPD)-anticoagulated human plasma frozen within 24 h was bought from the Finnish Red Cross Blood Service (Helsinki, Finland), which is provided as an anonymized blood product for research use only that constitutes non-identifiable human material under the Declaration of Helsinki. Vitreous humour for proteomics experiments was isolated from porcine eyes supplied by HKScan Finland Oyj (Forssa, Finland) and homogenized as described previously [29,31]. Triton X-100 was from MP Biomedicals (Santa Ana, CA, USA). Hyaluronic acid (HA; 8–15 kDa) and all other chemicals were bought from Sigma-Aldrich (St. Louis, MO, USA). Buffer and calcein solutions were made in advance of the liposome preparations. The buffer contained 20 mM 4-(2-hydroxyethyl)-1-piperazineethanesulfonic acid (HEPES) and 140 mM sodium chloride (NaCl) in purified water, and the calcein solution for release studies had 60 mM calcein and 29 mM NaCl in purified water, both adjusted to pH 7.4 with sodium hydroxide (NaOH). Microwell dishes were from MatTek Corporation (Ashland, MA, USA).

### 2.2. DSPE–HA Conjugate

The DSPE–HA conjugate was produced with the protocol described by Yao et al. with a few modifications [32]. Briefly, 50 mg of HA sodium salt (8–15 kDa) was added to 20 mL of purified water. The solution was dialyzed (2 kDa cut-off) against 1 L of 0.01 M hydrochloric acid (HCl) for 24 h to acidify the HA, and then against 1 L of purified water for 24 h. After the dialysis, the solution pH was increased to 9 with a few drops of 20% tetrabutylammonium hydroxide (TBA-OH). The mixture was then stirred for 2 h at room temperature and thereafter dialyzed against an excess of purified water. Water was removed from the final dialysis product by lyophilization with SCANVAC CoolSafe 110 (LaboGene ApS, Allerød, Denmark).

The freeze-dried HA–TBA powder was dissolved in 3.5 mL of anhydrous dimethyl sulfoxide (DMSO) under an argon atmosphere, and the solution was stirred at 60 °C until the powder dissolved completely. Then, 16.3 mg of DSPE and 6.1 μL of dry triethylamine were mixed with 0.87 mL of anhydrous chloroform, and the solution was sonicated for a short period until all the particles visibly dissolved. The HA and DSPE solutions were combined and stirred for 2 h at 60 °C. Thereafter, a solution containing 9.2 mg of sodium triacetoxyborohydride (NaBH(OAc)_3_) and 0.87 mL of anhydrous DMSO was added dropwise to the DSPE–HA–TBA solution. This mixture was allowed to react for 72 h at 60 °C and then cooled to room temperature.

Chloroform was evaporated from the suspension under a constant nitrogen gas flow. Then, 30 mL of purified water was added to the suspension, which was then centrifuged at 10,000× *g* for 30 min. The supernatant was dialyzed against 0.01 M HCl for 24 h and under purified water for an additional 24 h to remove the acidic form of the TBA salt. Finally, the product was freeze-dried with the SCANVAC CoolSafe 110 (LaboGene ApS) over the duration of two days.

### 2.3. DSPE–HA Conjugate Analysis

The conjugation products were analyzed with nuclear magnetic resonance spectroscopy (NMR) and Fourier-transform infrared spectroscopy (FT-IR). The ^1^H-NMR analysis was performed with a 400 MHz NMR Bruker AVANCE-III HD spectrometer (Bruker BioSpin, Rheinstetten, Germany). For the measurement, DSPE was dissolved in deuterated chloroform (CDCl_3_), whereas the DSPE–HA conjugate was dissolved in deuterated dimethyl sulfoxide (DMSO-*d_6_*). 

A Vertex 70 FT-IR spectrometer (Bruker Optics, Ettlingen, Germany) equipped with a MIRacle attenuated total reflectance attachment was used in the FT-IR analysis, so that a tiny amount of either DSPE or DSPE–HA conjugate was placed on a diamond single-bounce crystal (Pike Technologies, Fitchburg, WI, USA). The scan was performed in the 650–4000 cm^−1^ wavelength range, while the spectral resolution was 4 cm^−1^. The background signal measured from a blank crystal was deducted from the average of 256 scans with Opus software.

### 2.4. Light-Activated Liposomes

ICG liposomes were prepared via the thin film hydration method followed by an extrusion and purification, as described previously with a few modifications [20,29]. Three formulations were prepared, uncoated (F1), PEG-coated (F2), and HA-coated (F4) liposomes with a basic lipid composition of DPPC/DSPC/(18:0)LysoPC at molar ratios of 75/15/10, supplemented with either 4 mol DSPE, 4 mol DSPE-PEG2000, or 1 mol DSPE–HA. The molar ratio of the DSPE–HA was chosen to correspond with the coating polymer mass of traditional PEG-coated liposomes. The thin lipid layer was hydrated with 0.322 mg/mL of ICG in 500 μL of HEPES buffer or calcein solution. In the case of the HA-coated liposomes, DSPE–HA was added during the hydration step instead of the initial mixture in chloroform. Prior to the thin film hydration step, the hydration solution with DSPE–HA was vortexed and heated to 64 °C until the lipid conjugate dissolved. The liposomes were extruded 11 times at 60 °C through a polycarbonate membrane with 100 nm diameter pores (unless otherwise specified) with a syringe-type extrusion device (Avanti Polar Lipids). Thereafter, the liposomes were quickly cooled and stored in a refrigerator. The unencapsulated calcein and ICG were removed by gel filtration through a Sephadex G-50 (Sigma-Aldrich) column with HEPES buffer elution. The final lipid and ICG concentration of the purified samples was 1.5 mM and 30 μM, respectively. For protein corona studies, similar liposomes with PEG (F3) and HA coating (F5), but without ICG, were also prepared. Additionally, anionic 50 nm liposomes with a lipid composition of DPPC/DSPC/DSPG/(18:0)LysoPC/DSPE (F6) or DSPE-PEG2000 (F7) at molar ratios of 75/5/10/10/4 were prepared as described by Tavakoli et al. [29].

### 2.5. Size and Zeta Potential Analysis in Buffer

The hydrodynamic diameters of the liposomes were analyzed with a Zetasizer APS dynamic light scattering (DLS) automated plate sampler (Malvern Instruments, Malvern, United Kingdom) and reported as size distributions by particle number and polydispersity index (PdI). The zeta potential was measured with a Zetasizer ZS (Malvern Instruments) using DTS1070 cuvettes. All samples were diluted 1:10 (*v*/*v*) with buffer and measured in triplicate with 10 subruns at room temperature.

### 2.6. Size Analysis by Flow Cytometry in Vitreous and Plasma Samples

Flow cytometry (Apogee Flow Systems, Hertfordshire, UK) was used to differentiate between the liposomes and other possible porcine vitreous and human plasma components of the same size by large-angle light scattering (LALS) and by detecting the calcein fluorescence to separate the liposome population. The LALS measurements returned the relative size and coefficient of variation (CV) of the liposome population. A calibration line for the liposome size was produced by measuring liposomes of different size in buffer with DLS, as described earlier, and then measuring the same samples with Apogee flow cytometry (Appendix A). The generated calibration line had the following equation (*R^2^* = 0.96):(1)LALS=21.666×DLS(nm)−888.11. The flow cytometry signal from bulk vitreous and plasma without liposomes was analyzed separately, and only the liposome population was gated into the final data (Appendix A).

### 2.7. Differential Scanning Calorimetry

The phase transition temperature (T_m_) was analyzed using 20 μL of the unpurified sample with differential scanning calorimetry (Mettler Toledo DSC 823e, Mettler-Toledo GmbH, Greifensee, Switzerland). The sample and reference pans were heated using a linear temperature gradient from 25 °C to 80 °C in a nitrogen environment. The phase transitions were detected as negative endothermic peaks in the baseline-corrected thermographs following analysis with STARe software (Mettler Toledo).

### 2.8. ICG Stability

ICG degrades quickly in aqueous solutions and loses its absorbance peak at the 800 nm wavelength [33]. Thus, the absorbance can be used to measure ICG leakage from liposomes. The ICG amounts were measured in black clear-bottom well plates (Corning Inc., Corning, NY, USA) using a Varioskan LUX multimode plate reader (Thermo Fisher Scientific Inc., Waltham, MA, USA). The 100 μL sample absorbances were measured at 800 nm, and the amount of intact ICG (ICG_i_) compared to the initial concentration was calculated using Equation (2).
(2)$${\mathrm{ICG}}_{i}=\frac{{\mathrm{ICG}}_{t}}{{\mathrm{ICG}}_{0}}\times 100\%,$$
where ICG_t_ is the absorbance at the specific time point, and ICG_0_ is the absorbance at the start of the experiment.

### 2.9. Temperature-Induced Release

The release of calcein caused by heating was measured at 35, 37, 38, 39, 40, 41, 42, and 44 °C. Firstly, 490 μL of HEPES buffer was warmed to the desired temperature in an Eppendorf ThermoMixer C (Eppendorf AG, Hamburg, Germany). Once the target temperature was reached, 10 μL of the liposome solution was added into the pre-warmed buffer in the incubator, and the mixture was stirred for 10 min at 300 rpm. The sample was then rapidly cooled in a bucket of ice. The cold control that contained only 490 μL of buffer was kept in the fridge until the addition of 10 μL of the liposomes just before the measurements. Finally, the calcein fluorescence (excitation 493 nm, emission 518 nm) was measured with Varioskan LUX (Thermo Fisher Scientific). The calcein release (R) was calculated as percentages using Equation (3).
(3)$$R=\frac{F\;-{\;F}_{0}}{F_{100}\;-{\;F}_{0}}\times 100\%,$$
where F is the fluorescence of the sample, F_0_ denotes the background fluorescence measured from the control, and F_100_ is the maximum fluorescence when the content is completely released from the liposomes (addition of 10 µL of 1% Triton X-100 solution for total decomposition).

### 2.10. Light-Activated Release

Photothermal calcein release was triggered using a compact single-mode laser module ML6700 (Modulight Inc., Tampere, Finland). Then, 500 μL of liposomes were incubated in HEPES buffer (37 °C), porcine vitreous (35 °C), or human plasma (37 °C) samples in Eppendorf ThermoMixer C (Eppendorf AG). After a sufficient incubation period, the sample was exposed to 800 nm laser for 5 or 15 s with the intensity of 3.2 W/cm^2^, based on our previous studies [18,21]. During the light exposure, the control tube was shielded from the laser. Right after the laser irradiation, the sample and the control were immediately cooled down in ice. The calcein fluorescence was measured with Varioskan LUX as described earlier and the release was calculated using Equation (3).

### 2.11. Liposome Stability in Vitreous and Plasma

The storage stability of the PEG- and HA-coated liposomes was studied for one week in porcine vitreous and human plasma (25% liposome solution/75% biological fluid) samples. The mixtures were stored in the ThermoMixer C (Eppendorf AG) under 300 rpm shaking in tubes at 35 °C for the vitreous and 37 °C for the plasma samples, respectively. Samples were collected at time points of 0, 1, 3, 6, 48, 96, and 168 h, and their calcein leakage, ICG stability, and liposome size were determined as described above. Light-activated release was performed at the 3 h time point for vitreous (35 °C) and plasma (37 °C) samples, as well as at the 24 h time point for the vitreous samples.

### 2.12. Diffusion of Liposomes in Intact Porcine Vitreous

Vitreal diffusion of uncoated, PEG-coated, and HA-coated liposomes in the porcine eye was investigated as reported previously [29]. In brief, fresh porcine eyes (age: six months) enucleated with a transconjunctival incision 15 min after sacrificing the animal were obtained from the slaughterhouse (HKScan Finland Oyj). After cutting out the extraocular tissues, the eyes were shortly immersed in 70% ethanol and then kept in phosphate buffered saline (PBS) buffer at 4 °C. The anterior segment of the eye was cut circumferentially 2 mm below the limbus, and the lens was removed. Then, 50 µL of 0.25 mg/mL fluorescently labeled liposomes was injected into the center of the exposed vitreous humor at a depth of 0.5 cm using a 30 G insulin syringe (BD, Franklin Lakes, NJ, USA). Next, a 35 mm microwell dish with a 14 mm diameter glass window (MatTek Corporation) was placed on top of the cut eyecup, while avoiding air bubble formation between the glass window and the vitreous humor. In order to prepare the eye for imaging, the dish-covered eyecup was carefully flipped over, and the edges of the eye were fixed to the dish using Loctite Precision super glue (Henkel Corp., NJ, USA). Imaging was carried out at 37 °C with a spinning disc confocal microscope (Marianas, Intelligent Imaging Innovation (3i) Colorado, USA) and 50 ms temporal-resolution movies were recorded using Slidebook^®^ Software v. 6 (Intelligent Imaging Innovation). Movement of the Cy5-labeled liposomes was tracked using the single-particle tracking technique, and the trajectories were analyzed with Imaris v. 9.2 software (Bitplane AG, Zurich, Switzerland). Finally, the particle mean square displacement (MSD) was computed using the @msdanalyzer MATLAB plug-in (MathWorks Inc., Massachusetts, USA), and the vitreal diffusion coefficient of liposomes (D_v_) was calculated using Equation (4).
(4)$$D_{v}=\frac{\mathrm{MSD}}{2d\tau },$$
where MSD is derived from the slope of the MSD curve, *d* is defined as dimensionality of the movement (*d* = 2 for 2D tracks), and τ is the lag time of the calculated displacement. The results are reported as D_w_/D_v_, which indicates how many times more rapidly the liposomes diffuse in water compared to the more viscous vitreous gel. The theoretical diffusion in water (D_w_) was calculated according to the Stokes–Einstein equation at 37 °C, and the radii of liposome particles were obtained from DLS measurements as described above.

### 2.13. Protein Corona Structure and Composition

The two-wavelength surface plasmon resonance (SPR) measurements and calculations, LC–MS/MS data acquisition, and sample preparation were conducted as described previously [28,29]. The LC gradients used for human plasma and porcine vitreous samples were 1 h and 3 h, respectively. The 500 µL liposome injection volume per replicate corresponds to ~16 × 10^12^ particles, with ~13.2 × 10^9^ immobilized liposomes on the sensor with ~55 nm^2^ of active surface area. Each SPR replicate run accounts for the measured number of immobilized liposomes on the sensor before the determination of corona thicknesses, which assumes random sequential adsorption [28,29]. Raw data from earlier publications in human plasma with neutral 100 nm ICG liposomes (F1–F3) [28] were integrated with new data for 100 nm HA liposomes with and without ICG (F4–F5) and 50 nm ICG liposomes with and without PEG. Raw data from porcine vitreous studies with the 50 nm ICG liposomes (F6–F7) [29] were integrated with new data for the optimized pegylated formulation F2 and the HA liposome F4 in porcine vitreous humor. The non-functional liposomes without ICG were not tested in vitreous humor.

Upon integration of the two datasets, principal component analysis excluded the presence of significant batch effects. As such, LC–MS/MS raw data were processed and median-normalized without additional batch effect corrections. Protein identification and quantification of protein groups (LC–MS/MS data processing) were carried out with the MaxQuant v. 1.6.1.0 [34], with the UniProtKB FASTA file either for *Sus scrofa* (40,701 protein and 23,223 gene entries) or for *Homo sapiens* (86,725 protein and 20,605 gene entries), to which 245 commonly observed contaminants and all reverse sequences were added. Differential abundance and hierarchical clustering analyses for relative enrichment were carried out with Perseus v. 1.6.5.0 [35]. Protein identifications with non-zero intensity values in at least three samples in plasma and vitreous humor were retained for comparisons and annotated with pI, gene names, and sequences. Abundances were log_2_-transformed, and protein identifications were classified as “only identified by site”. Reverse sequences and potential contaminants were filtered out. When applicable, a protein’s median intensity (after normalization) across all samples where it was identified was used to rank and select high abundance (physiological expression or corona-enriched) proteins. Missing intensity values of unidentified protein groups were replaced with random numbers to mimic low abundance measurements, using the input from the normal distribution function in Perseus. After removal of outliers, 2–4 replicates of all formulations were retained for multiple sample testing.

ANOVA with a Benjamini–Hochberg false discovery rate (FDR) of 0.05 on the top 20 proteins from the assigned groups (source, HC, SC) was followed by two-sample Student’s *t*-tests with Benjamini–Hochberg FDR correction. To determine the protein physicochemical properties, gene names for the majority protein within each mass spec-identified protein group and their default protein sequences were obtained from UniProtKB. Unmapped identifications were excluded, and the same protein name was consistently retained for genes that encode multiple proteins. The same sequences were used to calculate the theoretical pI, molecular weight, and grand average of hydropathy (GRAVY) using the the ExPASy server ProtParam tool (https://web.expasy.org/protparam/) or the GRAVY Calculator (http://www.gravy-calculator.de/). The sums of the aromatic residues (phenylalanine, tyrosine, tryptophan, with and without histidine) were calculated from the sequences using the LEN function in MS Excel that returns the number of characters in a text string. For analysis of common gene subsets without accounting for enrichment, the HC Intersection in plasma and the HC Union in vitreous humor were used. Additional comparisons between gene sets were conducted using Venn diagram software (http://bioinformatics.psb.ugent.be/webtools/Venn/), followed by GeneMANIA (http://genemania.org) for functional analysis.

### 2.14. Statistical Analysis

All stability experiments were carried out in triplicate, and the data are presented as mean ± standard deviation (S.D.). The analysis consisted of a Student’s *t*-test to evaluate the significance between the groups (*p* < 0.05). Outliers were detected and excluded using Grubb’s outlier test (*p* < 0.05). Additional statistics were conducted with Prism 8.2.1 (GraphPad Software Inc., La Jolla, CA, USA).

## 3. Results

### 3.1. Analysis of the DSPE–HA Conjugate

The DSPE–HA conjugate was analyzed with ^1^H-NMR and FT-IR spectroscopy (Appendix A). The ^1^H-NMR spectra showed the typical peaks for DSPE–HA sugar rings (3–4 ppm), *N*-acetate (1.7–2 ppm), methylene (~1.2 ppm), and terminal methyl (~0.9 ppm) as reported by other groups [32,36]. DSPE and DSPE–HA were analyzed with FT-IR. The DSPE–HA spectra showed comparable peaks (3366, 2918, 2853, 1730, and 1643 cm^−1^) with both the HA and the DSPE spectra in accordance with a previous report by Yao et al. [32]. The analysis confirmed the successful production of the DSPE–HA conjugate.

### 3.2. Properties of the HA-Coated Liposomes

The physicochemical properties of the four different liposome formulations produced for the experiments are displayed in Table 1. The lipid composition of all liposomes is based on an optimized formulation for light-activated ICG liposomes reported earlier [18,20]. Depending on the liposome coating, the ICG is either located within the lipid bilayer or bound by the surface polymers [21]. HA-coated liposomes were produced at different sizes (~70 nm and ~100 nm), demonstrating their robust size control. Polydispersity indexes were in the range of 0.05–0.08. The HA-coated liposomes showed lower zeta potential than the uncoated and PEG-coated liposomes. The PEG- and HA-coated liposomes had comparable ICG absorbances, suggesting that the polymer coating did not affect ICG binding on the liposomes, but the uncoated liposomes bound less ICG (Table 1).

The coating had no significant effect on the lipid phase transition in the liposomes (Figure 1A). The transition to the liquid-disordered phase began at ~42 °C and reached its peak at ~44 °C. On the other hand, temperature-induced calcein release from small HA-coated liposomes took place at 39–41 °C, while the 100 nm liposomes released their contents at 38–40 °C (Figure 1B). Light activation in buffer for 15 s was sufficient to induce complete content release from all liposomes, while, at 5 s exposure, the small HA-coated liposomes showed lower content release than the 100 nm liposomes (Figure 1B). Differences in content release between light-activated and normal liposomes were significant (*p* < 0.01) (Figure 1C).

Liposomes with a doubled amount of HA (molar ratio = 2) did not show significant differences compared to the liposomes with lower HA content (Appendix A). In order to determine the ICG binding capacity of the HA coating, we successfully doubled the amount of ICG (Appendix A). The ICG was bound to the liposomes at higher quantities (ICG absorbance of 1.61 ± 0.04 a.u., as compared to 0.88 ± 0.12 a.u. for the regular HA-coated liposomes). However, if the ICG content was doubled at normal HA levels (molar ratio = 1), significant calcein leakage was observed below 37 °C, which rendered these liposomes unsuitable for in vivo use. Due to the suboptimal stability and pharmacokinetics of the uncoated liposomes, they were excluded from the stability studies.

### 3.3. Stability Studies 

#### 3.3.1. ICG Stability and Passive Content Leakage

Light-activated liposomes should stay intact and avoid passive content leakage for sufficient duration. The stability of the PEG- and HA-coated liposomes was assessed in porcine vitreous and human plasma samples at 35 °C and 37 °C, respectively (Figure 2). The ICG absorbance and calcein leakage were measured for one week. Significant changes in ICG absorbance were not seen in vitreous or plasma samples.

Both PEG- and HA-coated liposomes showed significant calcein leakage in 2–4 days in the vitreous sample (Figure 2B). Leakage of the liposomal contents was faster in plasma than in vitreous samples, as complete leakage was observed within 24 h (Figure 2E). However, the content release in plasma from the HA-coated liposomes was significantly slower than from the PEG-coated liposomes.

#### 3.3.2. Light-Activated Content Release in Vitreous and Plasma Samples

The light-triggered content release mechanism should be functional in biological media. Light-activated calcein release from HA-coated liposomes was studied at 3 h and 24 h in the vitreous (35 °C) and at 3 h in plasma (37 °C) samples (Figure 3). The longer light exposure (15 s vs. 5 s) significantly increased the calcein release in the vitreous sample at 3 h and 24 h (* *p* < 0.05). Even more pronounced triggered release was observed in plasma (Figure 3).

#### 3.3.3. Liposome Size Change

Flow cytometry was used to analyze the liposomal size change during the stability study (Section 3.3.1), because the large number of natural particles in biological solutions prevents reliable measurements with DLS. The large-angle light scattering (LALS) measurements of the liposome populations in vitreous and plasma samples with Apogee flow cytometry were converted into hydrodynamic diameters using a calibration curve (Figure 4B; Appendix A). Only the liposome population was gated based on the fluorescence, removing the naturally occurring particle aggregates from the signal (Appendix A). The coefficients of variance were 70–100 for all samples throughout the experiment without showing a clear tendency for increase or decrease. 

The diameter of the PEG-coated liposomes remained unchanged throughout the one-week exposure period in vitreous. In plasma, a gradual size increase up to a diameter of 155 nm was observed. The relative size increase was even higher for the HA-coated liposomes in plasma (270 nm at two days, 318 nm at one week), whereas, in the vitreous sample, the HA-coated liposomes reached a diameter of 150 nm in the beginning and remained at that level until the end of the experiment.

### 3.4. Protein Corona Formation

#### 3.4.1. Hard and Soft Corona Thickness

Surface plasmon resonance (SPR) measurements under dynamic conditions in undiluted human plasma and porcine vitreous samples were carried out to determine the hard (HC) and soft corona (SC) thicknesses and separate these fractions for proteomics analysis of their composition [28]. Due to the higher number of the samples as a result of data integration from earlier studies, additional formulation numbers (F1–7) and color codes based on charge and size are used (Figure 4A). In plasma, the HA liposomes formed a 5.4 ± 1.2 nm hard corona and a 9.3 ± 1.6 nm soft corona, which is on par with the 4.9 ± 0.7 and 10.6 ± 1.6 nm SC obtained for the PEG liposome (Figure 4C). The uncoated 50 nm anionic liposome without PEG (F6) formed a very thin HC and SC in a consistent trend with the uncoated 100 nm neutral liposome (F1). The HA liposome without ICG (F5) formed a thinner corona than its ICG-containing counterpart.

In the vitreous sample, the HA liposome formed a thicker 1.6 ± 0.3 nm HC and 2.7 ± 0.5 nm SC than the PEG liposome (1.1 ± 0.2 nm and 1.6 ± 0.3 nm, respectively). Both 100 nm formulations formed thinner coronas than the 50 nm anionic liposomes regardless of surface charge or polymer coating, and the SC thickness suggests that it comprised only a few additional loosely interacting proteins in addition to the HC. The protein concentrations of the undiluted porcine vitreous and human plasma samples determined using the bicinchonic acid (BCA) assay were 1.5 mg/mL and 75.8 mg/mL, respectively. These correspond to dynamic exposures of 0.25 mL at 375 µg of protein per replicate in the vitreous sample (1.4 pg/nm^2^/min) and at least 0.75 mL at 57 mg if protein per replicate in plasma (57 pg/nm^2^/min). The unfunctional formulations without ICG were not tested in the vitreous sample. 

#### 3.4.2. Hard and Soft Corona Composition

The HC and SC corona fractions from Figure 4C,D were eluted for high-resolution proteomics analysis with nanoliquid chromatography tandem mass spectrometry. Sufficient chromatographic separation was obtained with a one-hour linear gradient for plasma, while the vitreous samples required three-hour gradients. Figure 5 shows the relative enrichment and hierarchical clustering of the top 60 most abundant proteins in human plasma after median normalization. A list and a heatmap of the 178 proteins identified in at least three samples are provided in Appendix A.

The HC, SC, and source plasma cluster into distinct groups (Figure 5). In addition, the anionic and neutral liposome groups cluster regardless of liposome size. The notably enriched proteins in the *anionic* HC intersection are fibrinogen alpha and beta (FGA and FGB), complement component C3, thrombin (F2), and immunoglobulin variable regions (Figure 5A). *Neutral* liposomes were enriched with fibrinogen gamma (FGG), complement component C4, fibronectin (FN1), and apolipoproteins B (APOB) and E (APOE), as well as clusterin (CLU) (Figure 5B). The HA-coated liposomes were also enriched with the extracellular matrix component fibronectin (FN1) and the hyaluronic acid-binding proteins inter alpha trypsin inhibitors (ITIH2 and ITIH4) (dashed boxes in Figure 5). In addition, the HA-coated liposome with ICG (F4) was enriched with additional immunoglobulin light-chain regions, which was not observed in the absence of ICG (F5; Figure 5C). 

The vitreous corona samples had 728 proteins that were identified in at least three samples (Appendix A). The relative enrichment and hierarchical clustering of the top 60 most abundant proteins after median normalization are shown in Figure 6. In the vitreous sample, the corona subsections do not cluster based on charge or formulation. The relatively enriched HC proteins on the 50 nm anionic liposomes (F6–F7) and to a lesser extent on the 100 nm PEG-coated liposome (F2) include the cytoskeletal protein tubulin alpha (TUBA1B) and the glucose metabolism-associated glyceraldehyde-3-phosphate dehydrogenase (GAPDH) (Figure 6A). Enrichment of tubulin beta chain variants (P02554; A0A287A275) was more pronounced in the PEG-coated liposome HCs (Figure 6B). These formulations were also enriched with clusterin (CLU) and hemoglobin subunit zeta (HBZ) in their HCs in addition to beta-crystallin (CRYBB1), which were mostly depleted on the HA-coated liposome (F4). The enrichment of the Dickkopf Wnt signaling pathway inhibitor 3 (A0A286ZNN6; DKK3) and an uncharacterized Ig-like protein (A0A286ZJL9) was less pronounced in these groups (Figure 6A,B), both of which were depleted on the HA-coated liposome (Figure 6C). Although GAPDH and TUBA1B were also enriched on replicates of the HA-coated liposome, their variants GADPHS and tubulin alpha chain (TUBA1A) showed higher enrichment (Figure 6d). The 50 nm anionic liposomes were also enriched with the retinol-binding protein 3 (RBP3) and albumin (ALB) in their HCs, which were mostly abundant in the SCs of the other formulations. One biological replicate of the PEG-coated liposome was an outlier, with a high abundance of immunoglobulins (Figure 6b,e,f). Both the HA-coated (F4) and PEG-coated liposomes (F2) with ICG were enriched with A0A286ZTT9, which is a 14-kDa immunoglobulin subtype deleted from the UniProt repository as obsolete in December 2019 (Figure 6f). Many of the other protein identifications that were not mapped into genes in Figure 6 are also immunoglobulins. Some of these proteins were found highly enriched in the SC of the HA-coated liposome with ICG (F4) and in some individual SC replicates (Figure 6e). The commonality between the source controls with different treatments (undiluted, diluted, SPR instrument flow-through) was 70% for plasma and 42% for vitreous samples.

#### 3.4.3. Properties of the Corona-Enriched Proteins

The HC and SC subsections in plasma were enriched with 17–47 proteins per formulation, while some vitreous corona samples had up to 620 enriched proteins per formulation (log_2_ fold-change ≥ 1.00). Notably, the HA-coated formulation with ICG (F4) and the 50 nm PEG-liposome (F7) were enriched with only 17 proteins in plasma. The HA liposome with ICG also had the smallest number of enriched proteins in its vitreous HC (102). The smaller dynamic concentration range of proteins in the vitreous sample is reflected in the narrower range of fold-changes, suggesting that the abundances of adsorbed proteins are also lower in vitreous than in plasma samples (Appendix A). To ensure comparability between samples and biological matrices, the top 20 most abundant proteins after median normalization were selected for the analysis of protein physicochemical properties (Appendix A). Their protein sequences were used to calculate the grand average of hydropathy (GRAVY), the theoretical isoelectric point (pI), the molecular weight (MW), and the number of the aromatic residues phenylalanine, tyrosine, tryptophan, and histidine (Ar (+His)) for the top 20 corona proteins in the corona subsections and the plasma or vitreous sources. The results for HC are displayed in Figure 7, while the corresponding figure for SC is shown in Appendix A). Summary data for both corona subsections in plasma and vitreous samples are provided as Appendix A.

There are no statistically significant correlations between liposome size or surface charge and the protein physicochemical properties. However, the anionic formulations had individual hydrophobic proteins in their HCs in plasma. The contributors of these positive hydropathicity values in plasma are immunoglobulin heavy chain V-III regions BRO and TIL. The median molecular weight of HC proteins is ~50 kDa in both plasma and vitreous samples, while the corresponding means are ~87 kDa and ~54 kDa, respectively, with higher variance in plasma. The anionic 50 nm liposomes (F6–F7) did not bind proteins with molecular weights above 200 kDa in their plasma HCs. The trend with aromatic residues mirrors that of the molecular weight in plasma. In vitreous humor, the second quartile of the pI is above the pH, indicating a higher abundance of proteins with a positive charge, while the majority of proteins in plasma and vitreous samples hold a negative charge at physiological pH. The top-range high-molecular-weight protein on the HA-coated liposome with ICG (F4) in vitreous is fibronectin (FN1). The protein properties for the top 20 proteins in SCs largely mirror the plasma and vitreous sources (Appendix A).

#### 3.4.4. Common Subsets of Hard Corona Proteins in Plasma and Vitreous Samples

There were 22 common species among all the enriched proteins found in the plasma HCs of more than one liposome (Appendix A). Among the seven proteins enriched in the HCs of all formulations, in addition to FGA and FGB, C3, and immunoglobulin constant fragments, were serotransferrin (TF) and alpha-2-macroglobulin (A2M), an iron transfer protein and protease inhibitor that are both highly abundant in plasma. In vitreous humor, only three proteins were common to the four formulations, including two structural crystallins and lactotransferrin (LTF), which is involved in iron transfer and immune functions (Appendix A). Overall, the vitreous common subset included 14 proteins, such as glyceraldehyde-3-phosphate dehydrogenase (GADPH), clusterin (CLU), albumin (ALB), and alpha-enolase (ENO1), the latter being associated with glucose metabolism and a number of functions including coagulation and immune stimulation. 

Figure 8 displays the commonality between plasma and vitreous for the common subsets for the four liposomes tested in both biological matrices. It demonstrates that the liposomes interact with 30–34 common proteins across the matrices, 74% of which are common to all formulations tested in both plasma and vitreous samples. These 28 proteins are listed in Figure 8B along with the corresponding top enriched biological functions in Figure 8C. Plasma HC common subset proteins carry a negative charge at physiological pH and are dominantly hydrophilic, similar to the top 20 most abundant proteins (Appendix A). The same holds for the vitreous common subset, which was composed of smaller-molecular-weight proteins with fewer hydrophobic residues on average. Among both common subsets, the retinol-binding protein 3 (RBP3) in the vitreous subset was the only one with a positive hydropathicity value. It is a visual cycle glycoprotein with a significantly higher molecular weight at 135 kDa compared to the 49 kDa mean. Although it also had the lowest pI, it was highly enriched on the 50 nm anionic liposome HCs, despite being mostly present in the source and SCs of the other formulations (Figure 6).

All of the common subset proteins were also among the top 20 most abundant proteins in plasma and vitreous samples (Figure 9). A list of the proteins and their properties in plasma and vitreous is provided in Appendix A. There is a high overlap between HC and SC identifications (Figure 9). Interestingly, an analysis of the *outlier* proteins that were specific to HA-coated liposomes with ICG (F4) in plasma showed the enrichment of collagen binding and extracellular matrix organization, with 67% of the predicted network explained by physical interactions (Appendix A; Appendix A). The small anionic liposomes were enriched with glycerolipid metabolism and actin filament organization functions, whereas the top enriched functions on neutral liposomes in plasma were related to glucose metabolism and immune activation. Overall, the top 20 HC enrichment patterns in plasma are mainly associated with blood microparticle and plasma lipoprotein particle-related functions, suggesting that they are secreted proteins that interact with lipid membranes (Appendix A; Appendix A). While complement activation did not rank highly, many of the proteins present in the common subset are associated with blood microparticle or plasma lipoprotein particles, or both, in addition to their immune system role (e.g., clusterin, CLU). Many of these proteins are also physical interaction partners. Glycolysis, visual perception, and blood microparticle were among the highest ranked functions in vitreous humor (Appendix A).

### 3.5. Vitreal Mobility

The vitreal mobility of HA-coated liposomes was compared against their PEGylated counterparts (Table 2) [29]. While there were no significant differences in the mean track speeds between the PEG- and HA-coated liposomes, both formulations had higher overall speeds than the uncoated liposome (*p* < 0.05; Figure 10B). The vitreal diffusion coefficient (D_v_), determined based on the mean square displacement by single-particle tracking analysis, showed that the HA-coated liposome diffused 1.5 times less than the PEG-coated liposomes despite their negative surface charge (Figure 10A). The diffusivity of the HA-coated liposomes was reduced 20-fold compared to their theoretical diffusion in water (D_w_), indicating that they encounter more resistance in the vitreous humor compared to the PEGylated and uncoated liposomes (D_w_/D_v_ = 13 and 16, respectively). Nonetheless, the mobility hindrance observed with the HA-coated liposome was minimal, as evident by the representative Brownian trajectories (Figure 10C).

## 4. Discussion

DSPE was conjugated with HA in-house to prepare hyaluronated liposomes, since there was no pre-existing conjugate available commercially. The successful reductive amination method was based on the work of Yao et al. [32]. The HA-coating on the liposomes did not significantly change the size, ICG encapsulation efficiency, or phase transition temperature of the liposomes (Table 1; Figure 1A). In the presence of hydrophilic coatings such as PEG, the ICG is stabilized by polymer clusters on the external side of the lipid bilayer [21]. Doubling the amount of both HA and ICG doubled the encapsulated amount of ICG without affecting the bilayer stability, while doubling ICG alone destabilized the membrane, causing passive leakage at 37 °C. This is in accordance with our previous results with PEG-coated ICG liposomes and suggests that HA is at least as efficient as PEG in stabilizing ICG [20]. The change in the surface coating of the 100 nm liposomes did not affect the temperature-induced release of calcein, although the small liposomes (~70 nm) had slightly higher release temperatures compared to the larger HA-coated liposomes (Figure 1B). A similar observation was previously made with PEG-coated ICG liposomes, which may be caused by curvature-modulated phase separation in the lipid bilayer [18]. This observation is supported by the light-activated release study, where small (~70 nm) HA-coated liposomes released less of their contents than the larger ones in response to a 5 s light exposure (Figure 1C).

Notably, while light-activated drug release from 100 nm HA-coated liposomes was comparable to the PEGylated liposomes in buffer, plasma, and vitreous samples, the HA coating improved liposomal stability in plasma at 37 °C (Figure 2C,D). Vitreal stability was comparable for both formulations over the period of one week. Following a 3-h incubation period in vitreous and plasma samples, the 15 s light exposure increased release by as much as 44% and 69% compared to passive leakage. The light-activated release was more pronounced in plasma, possibly due to the lipid bilayer-destabilizing effect of the biocorona on thermosensitive liposomes [37]. Therefore, the better plasma stability of the HA-coating especially within the first 12 h permits longer delays between intravenous (i.v.) administration and light activation, which is a beneficial property from a therapeutic perspective. The HA coating also made the liposomes significantly anionic (−11 mV), caused by the deprotonation of disaccharide carboxyl groups in an aqueous environment [38]. The analysis of the top 20 most abundant corona protein properties showed that the HA-coated liposomes were predominantly enriched with proteins that carry a negative charge in both plasma and vitreous pH, suggesting that the liposome anionic charge is also retained after coronation (Figure 7). This may partly explain the improvement in the stealth properties of HA-coated liposomes compared to their PEGylated counterparts [13,39].

In human plasma, ~5 nm HCs and ~10 nm SCs formed on the HA and PEG-coated liposomes, which correspond to sparse monolayers [28,40]. The formulation-dependent compositions also clustered based on liposomal surface charge (Figure 5). The influence of nanoparticle surface charge on the HC composition and the enrichment of fibrinogens under in vitro conditions were reported [41,42]. ICG also influences corona composition, as evidenced by the higher relative enrichment of complement and immunoglobulin proteins on the 8–15 kDa HA-coated liposome with ICG (F4) than without ICG (F5). This is likely due to trace endotoxins, since <400-kDa HA is not immunogenic [43]. The enrichment of immunoglobulin variable regions may be due to the particular epitope repertoire of the plasma donors [44].

In undiluted vitreous, the thinner ~2 nm HCs and ~3 nm SCs had formulation-independent compositions (Figure 4C,D). The slightly higher SC thicknesses but significantly different composition compared to the HCs suggest that it consists of only a few loosely interacting molecules. Interestingly, the 100 nm HA-coated formulation bound more proteins than the PEG-coated liposome in vitreous humor, while the reverse was seen in plasma. Interactions with the structural HA–collagen meshwork of the eye involve alpha and beta-crystallins, both abundant in vitreal coronas, as well as fibronectin and the inter-alpha trypsin inhibitors found in both plasma and vitreous HCs (Figure 8B). In plasma, the proteins that were only found on the HA-coated liposomes interact with fibrillar collagen type I and cell-adhesion protein collagen type XIV (Appendix A; Appendix A). These functions were not enriched in vitreous humor, possibly due to the low number of identified proteins on the HA-coated liposomes (Appendix A, Appendix A). Therefore, the “biological identity” ensuing from the HA coating may result in physical interactions with the structural collagen of the eye [45].

Consequently, these interactions may explain the slightly slower vitreal mobility of the HA-coated liposomes compared to PEG-coated liposomes. Since the increase in liposome diameter as a result of corona formation did not influence the diffusion of 50 nm anionic pegylated liposomes [29], it is inconsequential for neutral 100 nm liposomes at similar corona thicknesses. Although anionic nanoparticles diffuse freely in the vitreous humor [46], PEG-coated nanoparticles diffuse faster, possibly as a result of a “shielding” effect against charge-mediated interactions with the meshwork [47]. The HA- and PEG-coated liposomes diffused at comparable speeds, but significantly faster than the uncoated formulation (Figure 10). The random walk diffusion patterns indicate that particles with higher speeds do not necessarily relocate longer distances (Figure 10C). This is evident based on the comparable diffusion coefficients of the HA-coated and uncoated liposomes, despite the faster mean track speed of HA-coated liposomes. The more heterogeneous track speed distribution of the HA-coated liposomes may be explained by vitreous meshwork interactions.

The HA-coated liposomes had a low number of identified proteins with a narrow range of fold-changes in vitreous humor (Appendix A). However, as a result of median normalization, proteins with high abundances in individual replicates could be identified. The consistency (36–38% of the highest number of identifications) suggests that this is a consequence of physical interactions, such as protein complex binding to the liposomes. Stable physical interactions between proteins are needed for cellular processes, and evolutionarily conserved soluble complexes are abundant in the human proteome [48]. The vitreous humor is abundant in energy metabolism-related proteins, such as GAPDH, which interact with other proteins to exert their functions [49]. Earlier, we found that 20–22% of the highly enriched vitreal corona proteins on 50 nm anionic liposomes were physical interaction partners of GAPDH [29]. These showed a preference for the PEGylated formulation, as can also be seen in Figure 6A (F7). It is, therefore, possible that some of the less enriched proteins interact with GAPDH or other HC proteins instead of the nanoparticle surface, as demonstrated with complement protein C3 [50]. Protein complexes should be considered a factor that influences corona composition and structure.

It was proposed that protein complexity of the HC increases with nanoparticle size, and small nanoparticles bind proteins with more aromatic residues [51]. In our study, the 50 nm liposomes did not bind proteins above 200 kDa, which may reflect the effect of surface curvature, but this also decreased the number of aromatic residues. In vitreous humor, the uncoated anionic 50 nm liposome (F6) had the most distinct HC (50% unique protein identifications), and the small anionic liposome group had slightly more outliers than the other groups in plasma and vitreous samples (Figure 9). Therefore, our findings suggest an inverse relationship with HC complexity and liposome size. In addition, the properties of the top 20 most abundant proteins in HC or SC did not correlate with liposome size or charge (Figure 7 and Appendix A). As expected based on earlier studies [29,30], there was also no correlation with liposome properties and the corona composition in vitreous, although these liposomes had formulation-dependent coronas in plasma. However, our analysis confirmed that the HC and SC proteins are predominantly hydrophilic and carry an anionic charge in both plasma and vitreous samples. Similar observations were made in diluted plasma [42], as well as in diluted and undiluted vitreous humor with more limited sample sets [29,30]. Our results indicate that this trend is more universal than previously known.

Biocorona formation is influenced by nanoparticle-related and experimental factors [27,44] together with the biological environment, such as the nature and amount of its constituents and ionic strength [52]. However, the volume available to proteins influences all aspects of their behavior, including adsorption processes [53]. As a result, incubation in diluted and undiluted solutions results in different corona compositions and thicknesses [54]. Except for Digiacomo et al. [54], molecular crowding received little attention in the context of biocorona formation. Since there is a significant difference in plasma (~76 mg/mL) and vitreous (~1.5 mg/mL) protein concentrations, crowding agents such as structural HA are important modulators of ocular protein interactions [52,53,55]. Molecular crowding directly influences thermodynamics and diffusion, the drivers of protein adsorption and corona formation [56,57]. It is, therefore, likely that the differences in corona formation between plasma and vitreous samples, at physiological concentrations due to the use of undiluted fluids, are explained partly by structural HA. This would also explain the need for longer LC separation gradients and the lower number of protein identifications on HA-coated liposomes in vitreous humor. It should be noted that porcine vitreous humor is the best substitute to human vitreous humor in terms of its protein and HA concentrations [29] and viscoelastic properties [58].

The most abundant corona proteins most likely determine the ensuing biological responses, due to the higher probability of encountering cellular receptors in the correct orientation [26,59]. It is noteworthy that there is a common subset of 30–34 proteins in both plasma and vitreous samples, 74% of which were found in all of the HCs, including on 100 nm PEG-coated and HA-coated liposomes and on 50 nm uncoated and PEG-coated liposomes (Figure 8). Functional analysis showed an association with blood microparticle and immune activation functions (Figure 8C), which are also among the enriched functions for the top 20 HC proteins in plasma (Appendix A; Appendix A). Considering that nanoparticle aggregation and protein adsorption are driven by surface free energy, it is interesting to note the “double role” of lung surfactant proteins, where they participate in immune surveillance in addition to reducing alveolar surface tension [52]. The common subset, therefore, labels the liposomes “non-self” lipid particles regardless of the biological environment (plasma or vitreous), a possible consequence of combined surfactant and immune activity in response to the “abnormally” high surface energy of cell-mimetic lipid bilayers. This may also explain why the lipid corona is insensitive to nanoparticle properties [60,61].

The two-fold increases in hydrodynamic diameters measured with LALS results indicate aggregation (Figure 4B) [57]. Although the HA-coated liposomes showed comparable or improved stability in vitreous and plasma samples, respectively (Figure 2), they may be more susceptible to clustering and aggregation than the PEG-coated liposomes. This may also be influenced by the higher abundance of immunoglobulins [52,62]. However, protein adsorption did not influence the liposome stability or mobility in vitreous humor, regardless of the formulation. Future studies should explore the mechanism of biocorona formation in the ocular environment together with the ensuing biological responses, including elimination by ocular phagocytes and retinal cell uptake [63].

## 5. Conclusions

A novel HA–lipid conjugate was successfully synthesized for the preparation of light-activated indocyanine green liposomes. HA-coated liposomes were functional in light-triggered drug release in vitreous and plasma samples. The HA-coated liposomes showed adequate stability and mobility in the vitreous humor. Protein corona formation on the HA-coated liposomes was characterized in the vitreous and plasma samples, and its properties were compared to PEG-coated and uncoated liposomes. Suitable stability and pharmacokinetic properties are vital for triggered release delivery systems that rely on controlled phase changes. The results showed the significance of nanoparticle coating materials on the stability, as well as on the interactions with vitreal and plasma components. These in turn may affect the immunological reactions to the drug delivery system, the implications of which are essential for intravitreal therapy due to the immunoprivilege of the eye. Future studies should further elucidate the mechanisms and consequences of corona formation in vitreous humor, taking into consideration the molecular crowding activity of structural HA. In conclusion, the HA-coated light-activated liposomes are a promising drug delivery system for intravenous and ocular applications.

## Figures and Tables

**Figure 1 pharmaceutics-12-00763-f001:**
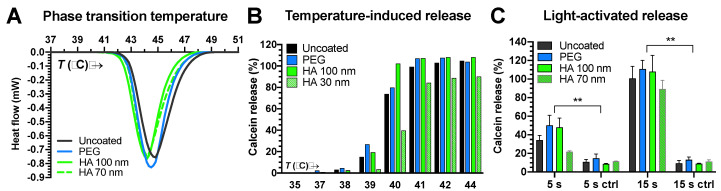
(**A**) Calorimetric analysis of liposomes. Heat flow as a function of temperature (*T*) is shown. (**B**) Calcein release from liposomes at different temperatures. (**C**) Light-activated content release from liposomes after 5 s or 15 s exposures (800 nm laser at 3.2 W/cm^2^) was significant compared to controls (ctrl) absent the light exposure (** *p* < 0.01).

**Figure 2 pharmaceutics-12-00763-f002:**
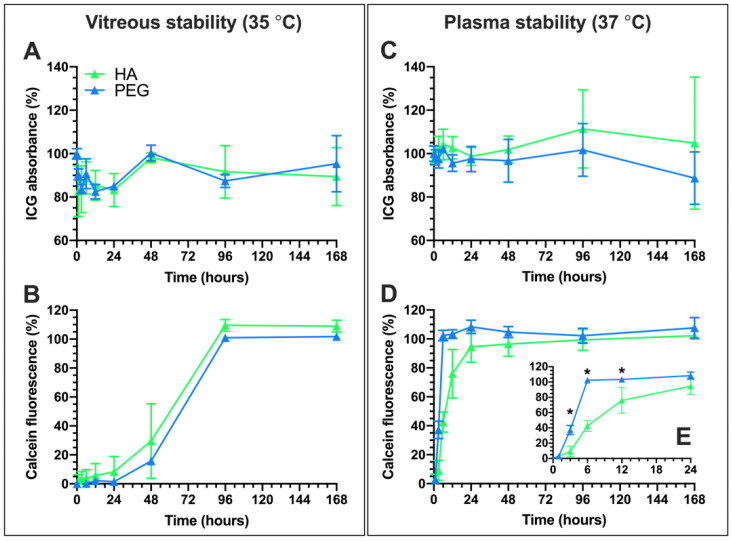
Stability of the HA- and PEG-coated liposomes at physiological temperatures monitored for 168 h (one week) based on ICG absorbance (**A**,**C**) and calcein fluorescence (**B**,**D**) in porcine vitreous (**A**,**B**) and human plasma (**B**,**D**) samples. Liposome solution (25%) was mixed with the biological fluids (75%). The snapshot (**E**) shows the time-points during the first 12 h of calcein release (* *p* < 0.05).

**Figure 3 pharmaceutics-12-00763-f003:**
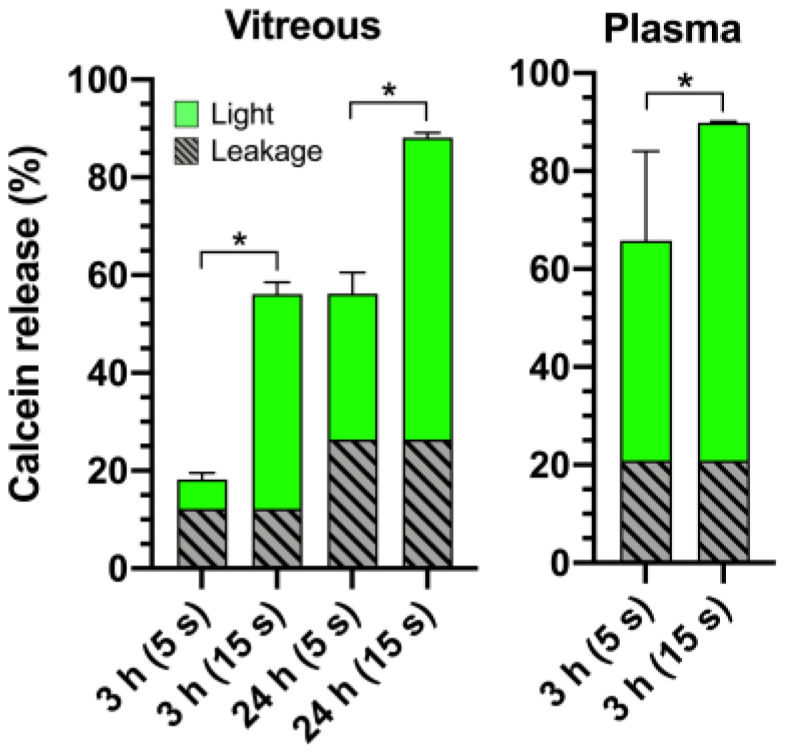
Light-activated release from HA-coated liposomes with 5 s and 15 s light exposures (800 nm laser at 3.2 W/cm^2^) and passive content release (leakage) following a 3 h or 24 h incubation of the liposomes in 75% porcine vitreous (35 °C) or human plasma (37 °C) samples. Significant differences in calcein release were observed between different exposure times (**p* < 0.05).

**Figure 4 pharmaceutics-12-00763-f004:**
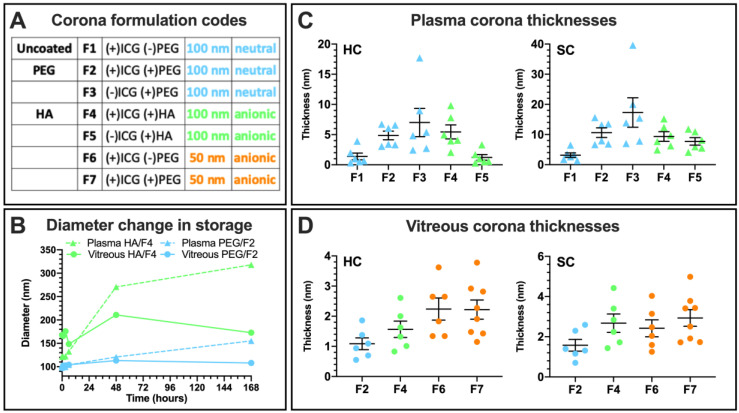
(**A**) Liposome formulations and their color-coded groups in corona experiments, combined with data from earlier studies [28,29]. (**B**) Large-angle light scattering measurements (LALS) for size change during storage in plasma and vitreous samples. (**C**) Surface plasmon resonance (SPR) measurements for hard (HC) and soft corona (SC) thickness in undiluted plasma with individual data points, mean, and SD. (**D**) Corona thicknesses in undiluted vitreous.

**Figure 5 pharmaceutics-12-00763-f005:**
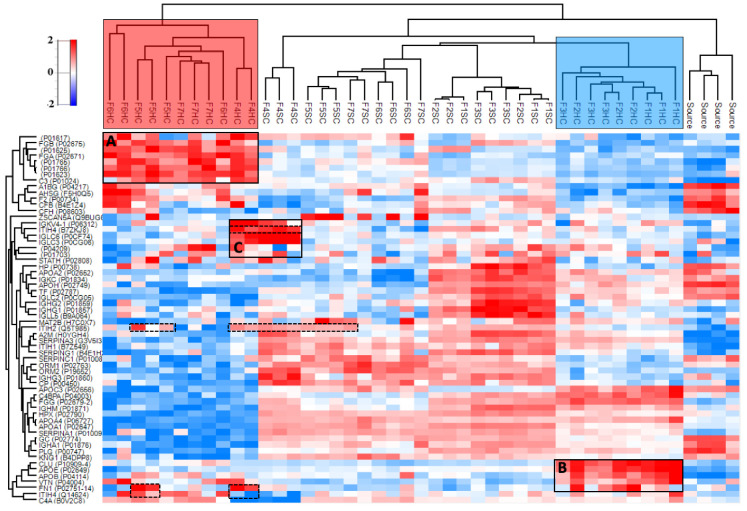
Heatmap of the top 60 most abundant proteins in at least three samples in human plasma after median normalization (178 proteins). Distinct clusters are observed for the HC, SC, and plasma source (including 1× and 7× diluted, surface plasmon resonance (SPR) instrument flow-through, and untreated). The HC subsections of the tested formulations cluster based on their anionic (*light-red box*) or neutral (*light-blue box*) surface charge. The corresponding significantly enriched proteins are highlighted with black boxes (**A**–**C**). The *dashed boxes* indicate the enrichment of hyaluronic acid-binding inter-alpha-trypsin inhibitor heavy chain proteins (ITIH2, ITIH4) and fibronectin (FN1). The range for relative enrichment (*red*) and depletion (*blue*) is two standard deviations from the mean on the log_2_ scale.

**Figure 6 pharmaceutics-12-00763-f006:**
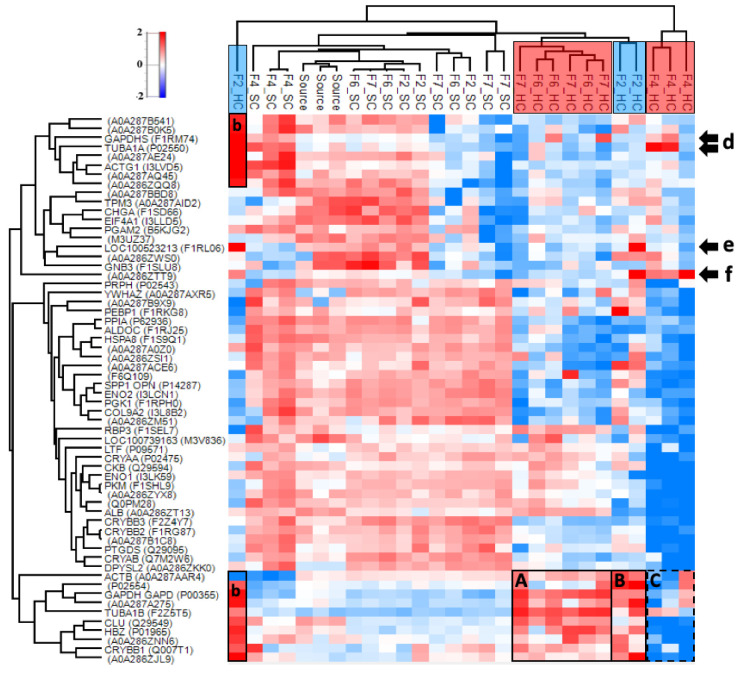
Heatmap of the top 60 most abundant proteins in porcine vitreous humor in at least three samples after median normalization (728 proteins). Distinct clusters are observed for the HC, SC, and plasma source. The HC subsections of the tested formulations do not cluster based on their anionic or neutral surface charge. The boxes (**A**–**B**) indicate the enrichment of tubulins, glyceraldehyde phosphate dehydrogenase (GAPDH), hemoglobin (HBZ), Dickkopf-related protein 3 (A0A286ZNN6), and clusterin (CLU) on the 50 nm anionic and 100 nm PEG-coated liposomes, which were depleted or less enriched on the HA-coated liposome (dashed box (**C**)). Interestingly, variants of glyceraldehyde phosphate dehydrogenase (GAPDHS) and tubulin (TUBA1A) were enriched on the HA-coated liposomes (**d**). One replicate of the PEG-coated 100-nm liposome showed high enrichment of these variants along with immunoglobulin family proteins (boxes (**b**,**e**)), which also enriched on the HA-coated liposome (**d**,**f**). An obsolete immunoglobulin subtype was enriched on the HA-coated liposome (**e**). The range for relative enrichment (*red*) and depletion (*blue*) is two standard deviations from the mean on the log_2_ scale.

**Figure 7 pharmaceutics-12-00763-f007:**
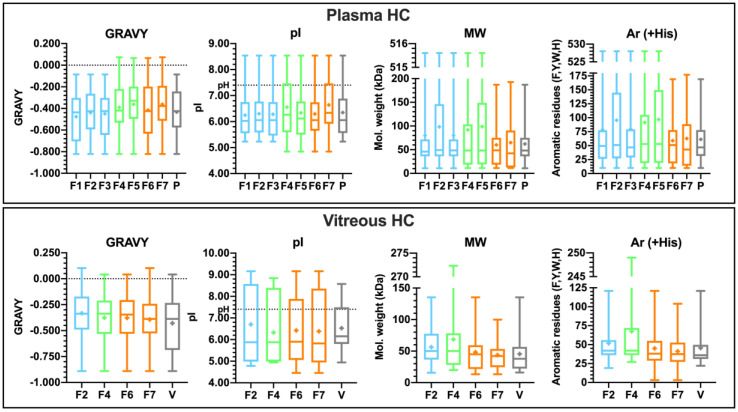
Properties of the top 20 enriched proteins in the hard coronas (HC) and the plasma or vitreous sources (P or V) with the max–min range, first and second quartiles, median, and mean (+). The color-coded groups are neutral 100 nm (*blue*), anionic 100 nm HA-coated (*green*), and small 50 nm anionic (*orange*) liposomes. GRAVY: grand average of hydropathy (positive score indicates hydrophobicity); pI: theoretical isoelectric point (pI under pH indicates net negative charge); MW: molecular weight; Ar (+His): aromatic residues phenylalanine, tyrosine, tryptophan, and histidine. Note the different *y*-axis ranges.

**Figure 8 pharmaceutics-12-00763-f008:**
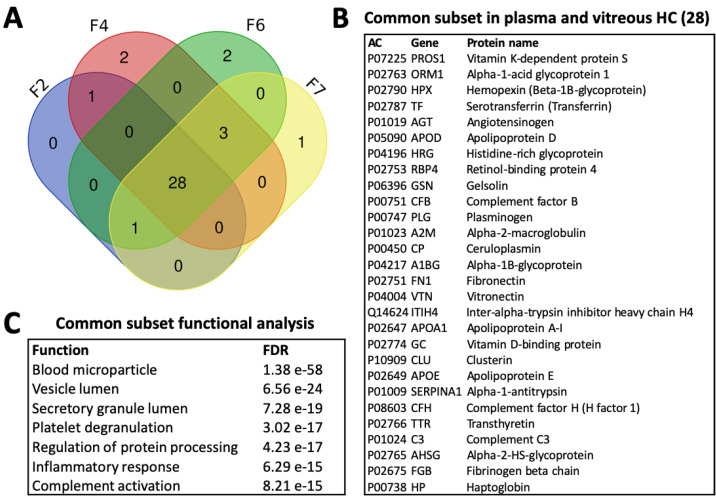
(**A**) Venn diagram of the commonalities between all proteins identified in the plasma and vitreous HCs of liposome formulations F2, F4, F6, and F7 without accounting for relative enrichment. (**B**) Proteins of the common subset. (**C**) Enriched biological functions in the common subset. AC: protein accession code; FDR: false discovery rate.

**Figure 9 pharmaceutics-12-00763-f009:**
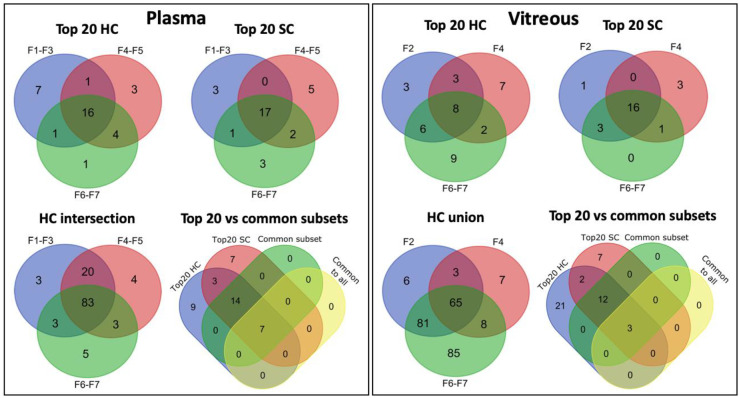
Venn diagrams of the commonalities between the neutral 100 nm (F1–F3 or F2), anionic 100 nm HA-coated (F4–F5 or F4), and anionic 50 nm (F6–F7) groups. Plasma: the top 20 most abundant proteins in the plasma HC (Top 20 HC) and SC (Top 20 SC), and their comparison with the common subsets (Top 20 vs common subsets), and for all HC proteins in found in plasma samples (HC intersection). Vitreous: the top 20 most abundant proteins in the vitreous HC (Top 20 HC) and SC (Top 20 SC), and their comparison with the common subsets (Top 20 vs common subsets), and for all HC proteins in found in vitreous samples (HC union).

**Figure 10 pharmaceutics-12-00763-f010:**
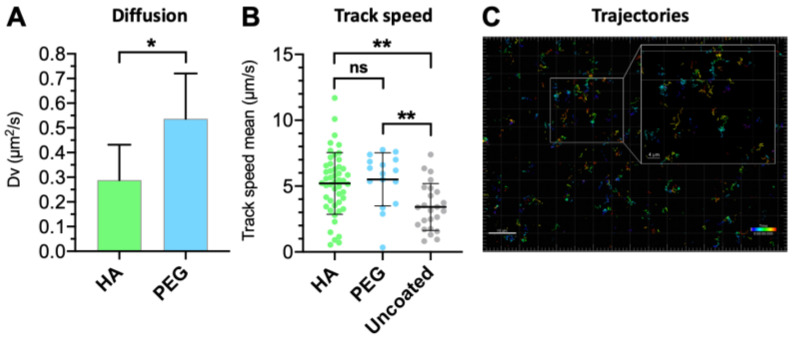
Comparison between (**A**) vitreal diffusion of HA- and PEG-coated light-triggered liposomes, analyzed by unpaired *t*-test (* *p* < 0.05); (**B**) ensemble-average speed of HA-coated, PEGylated, and uncoated light-activated liposomes (τ = 1 s). Means with significant differences are shown as analyzed by ordinary one-way ANOVA with Tukey’s multiple comparison test (** *p* < 0.01). Error bars indicate SD. (**C**) Representative trajectories of the HA-coated liposome in the intact porcine vitreous. Inset shows the zoomed-in view of tracks obtained using the single-particle tracking technique.

**Table 1 pharmaceutics-12-00763-t001:** Physico-chemical properties of the liposome formulations produced for the experiments (mean of triplicate measurements with SD). ICG—indocyanine green; PEG—polyethylene glycol; HA—hyaluronic acid; n.a.—not available.

	Size (nm)	Zeta Potential (mV)	ICG Absorbance (a.u.)
Uncoated	102 ± 24	−0.44 ± 1.32	0.72 ± 0.08
PEG	119 ± 30	−2.94 ± 0.69	0.90 ± 0.13
HA	104 ± 25	−11.27 ± 0.21	0.88 ± 0.12
HA (30 nm extrusion)	68 ± 16	n.a.	0.86 ± 0.03

**Table 2 pharmaceutics-12-00763-t002:** Comparison of the vitreal diffusion of HA-coated, PEG-coated, and uncoated liposome formulations. D_v_ was obtained experimentally by single-particle tracking for 1 s, and the D_w_ was calculated based on the Stoke–Einstein equation (*N* = 3). * Data from Reference [29].

Formulation	Mean Diameter (nm)	PdI	D_v_ (μm^2^/s)	D_w_ (μm^2^/s)	D_w_/D_v_
HA	116.1	0.008	0.28 ± 0.14	5.68	20.2
PEG *	107.6	0.086	0.47 ± 0.25	6.12	13.0
Uncoated *	125.8	0.035	0.33 ± 0.17	5.27	16.0

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
