# Peer review of "Light-Activated Liposomes Coated with Hyaluronic Acid as a Potential Drug Delivery System"

_pharmaceutics, 2020, doi:10.3390/pharmaceutics12080763_

Round 1

Reviewer 1 Report

Hyaluronic acid (HA)-coated liposomes of indocyanine green were investigated as a novel light-activated system for intravitreal delivery.  Protein corona formation, stability, and release property were characterized in the vitreous.  The data in different groups, such as uncoated, PEG-coated, HA-coated in the vitreous and in the plasma, were compared.  The results suggested that the HA-coated light-activated liposomes were functional in the vitreous and had a potential to allow site- and time-specific drug delivery.  The data are well-presented and discussed.  The research would provide new insights into light-activated systems for drug delivery.  Specific comments:

  1. The title of the manuscript suggests that the research would be for intravitreal drug delivery. The statements in Section 1 Introduction provided the objective of this research was to produce ocular liposomes (lines 79-83).  However, in the abstract and main text, the experiments were also conducted in the plasma.  The data in the vitreous and in the plasma were compared and significant discussions were on the comparisons.  It was also stated that the liposomes are promising alternative for intravenous and ocular drug delivery (lines 38 and 700).  Please clarify the research in this manuscript was for which route of administration. 
  2. The manuscript presents a very large amount of data and is relatively lengthy and less concise. If the research in this manuscript would be focused on intravitreal delivery, it would be suggested to remove the data obtained in the plasma or move them to the supplementary section. 
  3. Please add statements on the implications of the findings from this research. How the findings would be applied to improve intravitreal drug delivery?
  4. Formulation numbers were defined in the cited references. It may be considered to add formulation compositions and codes to the manuscript for readers’ convenience.   
  5. Please add more information to Figure 6 caption (e.g., A, B, C, b, d, e, and f).

Author Response

Thank you for the reviewers’ valuable comments. We have now modified the manuscript based on their advice.  Below you will find the answers to the comments point-by-point and we have indicated the changes with track-changes function as well as highlighting in the manuscript.

We appreciate the valuable input from the reviewers and feel that they have helped to strengthen and improve the manuscript considerably.

Reply to comment 1:

Since the manuscript demonstrates the suitability of the light-activated liposomes for both intravenous and ocular drug delivery, we agree with the reviewer that this should also be reflected in the title. The title was shortened accordingly to “Light-activated liposomes coated with hyaluronic acid as a potential drug delivery system”. We feel that this better represents the broad interests of the special issue.

Reply to comment 2:

Since we find that the plasma data is valuable addition to the manuscript and valuable to readers, as it shows the suitability of the light-activated liposomes for intravenous delivery, we amended the title of manuscript as per the reviewer’s earlier recommendation. The plasma data remains to be presented in the main text to better showcase these comparisons.

Reply to comment 3:

We appreciate the valuable opinion and agree that impact of work should be briefly addressed. Three sentences about the implications of the study have been added to the conclusions section.

Reply to comment 4:

As suggested by the reviewer, formulation numbers (F1-F7) with full detailed lipid compositions have been added to the materials and methods section. We hope this improves the convenience for the readers.

Reply to comment 5:

The figure caption was amended with information about the highlights specified by the reviewer. We hope this improves the clarity.

Reviewer 2 Report

In general through all text please add the meaning of the abbreviations (e.g. DSPE; TBA-OH) on the first time it appears on the text.

Figure 1b is accumulated released? Moreover the release is done at discrete values of temperature and not a continuous function of temperature as data is presented in figure?

To better characterization of the corona formations data from zeta potential before and after protein adsorption needs to be presented and this would give also some information as different proteins based on their isioelectric point would be more or less contributing for the corona in PEG and HA liposome’s.

Author Response

Thank you for the reviewers’ valuable comments. We have now modified the manuscript based on their advice.  Below you will find the answers to the comments point-by-point and we have indicated the changes with track-changes function as well as highlighting in the manuscript.

We appreciate the valuable input from the reviewers and feel that they have helped to strengthen and improve the manuscript considerably.

Specific replies to the comments are:

"In general through all text please add the meaning of the abbreviations (e.g. DSPE; TBA-OH) on the first time it appears on the text."

We appreciate the attention to detail of the reviewer. The meanings for missing abbreviations of reagents used in the study were added in the materials and methods section.

"Figure 1b is accumulated released? Moreover the release is done at discrete values of temperature and not a continuous function of temperature as data is presented in figure?"

Figure 1B does not represent accumulated release. As detailed in methods section, the experiment measures the contents release in discrete temperatures. We fully agree with the reviewer that line graph in confusing in this case. For clarity, the format of Figure 1b was changed into a bar chart, since our intention is to display the results for non-accumulated release of calcein measured at discrete temperatures. The figure caption was also modified to make this more understandable.

"To better characterization of the corona formations data from zeta potential before and after protein adsorption needs to be presented and this would give also some information as different proteins based on their isioelectric point would be more or less contributing for the corona in PEG and HA liposome’s."

In the manuscript, an in situ method was used to study protein corona formation under dynamic conditions in undiluted vitreous and plasma, which cannot be used to determine the zeta potential directly. Instead, the information on the expected charge was based on the isoelectric points of the corona proteins identified in the proteomics analysis, which were calculated from their amino acid compositions to ensure comparability. The isoelectric points of the proteins are presented along with other details in Supplementary File 3-6 and per liposome formulation in Fig. 7 and Supplementary Figure S7. No correlations were found between the protein isoelectric points and the properties of the liposomes in our study.

It is currently not possible to accurately measure the zeta potential following protein adsorption on nanoparticles, since it is carried out in buffer and sample preparation for zeta potential measurements influences the composition of the protein hard corona and eliminates the soft corona entirely. As a result, conducting these studies separately would not produce data that is comparable with the other results presented in this manuscript, such as the protein corona compositions and thicknesses.

Round 2

Reviewer 1 Report

n/a